# Immunomodulatory Effects of Edible and Medicinal Mushrooms and Their Bioactive Immunoregulatory Products

**DOI:** 10.3390/jof6040269

**Published:** 2020-11-08

**Authors:** Shuang Zhao, Qi Gao, Chengbo Rong, Shouxian Wang, Zhekun Zhao, Yu Liu, Jianping Xu

**Affiliations:** 1Institute of Plant and Environment Protection, Beijing Academy of Agriculture and Forestry Sciences, Beijing 100097, China; zhaoshuang@ipepbaafs.cn (S.Z.); gaoqi@ipepbaafs.cn (Q.G.); rongchengbo@ipepbaafs.cn (C.R.); wangshouxian@ipepbaafs.cn (S.W.); zhaozhekun@ipepbaafs.cn (Z.Z.); liuyu@ipepbaafs.cn (Y.L.); 2College of Life Sciences and Food Engineering, Hebei University of Engineering, Handan 056038, China; 3Department of Biology, McMaster University, Hamilton, ON L8S 4K1, Canada

**Keywords:** bioactive compounds, FIP, human health, immunomodulation, induced apoptosis, lectin, medicinal mushrooms, polysaccharide, terpenes and terpenoids

## Abstract

Mushrooms have been valued as food and health supplements by humans for centuries. They are rich in dietary fiber, essential amino acids, minerals, and many bioactive compounds, especially those related to human immune system functions. Mushrooms contain diverse immunoregulatory compounds such as terpenes and terpenoids, lectins, fungal immunomodulatory proteins (FIPs) and polysaccharides. The distributions of these compounds differ among mushroom species and their potent immune modulation activities vary depending on their core structures and fraction composition chemical modifications. Here we review the current status of clinical studies on immunomodulatory activities of mushrooms and mushroom products. The potential mechanisms for their activities both in vitro and in vivo were summarized. We describe the approaches that have been used in the development and application of bioactive compounds extracted from mushrooms. These developments have led to the commercialization of a large number of mushroom products. Finally, we discuss the problems in pharmacological applications of mushrooms and mushroom products and highlight a few areas that should be improved before immunomodulatory compounds from mushrooms can be widely used as therapeutic agents.

## 1. Introduction

In clinical practice, immunomodulators are usually classified into three categories: immunosuppressants, immunostimulants, and immunoadjuvants [1]. Their market share has increased rapidly over the past few years due to wide-ranging medical applications for patients that require human immune system modulations. Immune system modulations are also commonly used as prophylactic medicine for an increasing number of healthy people [2,3]. While most immunomodulators are synthetic or semi-synthetic compounds, there has been a growing interest in natural immunomodulators. Many natural compounds have shown significant immunomodulatory and overall health-benefiting effects to humans, with no or minimal toxicity. These natural-based products with potential pharmacological and beneficial effects are increasingly perceived as safer than synthetic compounds by the general public [4,5]. Indeed, many of the currently used chemical drugs have negative side effects and the market share of natural immunomodulators is increasing rapidly with an annual growth rate of 8.6% [1,6].

Medicinal mushrooms (MMs) are an important source of natural immunomodulators. Representing a subset of all mushrooms, MMs are broadly defined as macroscopic fungi that are used in the form of extracts or whole mushroom powder for human health benefits [7]. The health benefits may come in the form of helping to prevent and/or treat diseases in humans, and to create a dietary balance of a healthy diet. Dating back to thousands of years, MMs have been historically used as herbal medicines for human health, such as for the treatment of infectious diseases, gastrointestinal disorders and asthmatic conditions [8]. The biomass or specific extracts from all developmental stages of MMs, including the fruiting body, sclerotium, mycelium, and spores, have been used as health foods or dietary supplements [9,10]. Some of the extracted nutrients from mushrooms known as mushroom nutraceuticals have been made into capsules or tablets as dietary supplements. Regular intake of these nutraceuticals has been associated with enhancement of the human immune response, leading to increased resistance to infectious diseases and faster recovery from a diversity of diseases [11].

At present, thousands of branded MM products are sold all over the world. The health benefits of MM products include anticancer, immune-stimulation, antioxidant, antihyperglycemic, antihypertensive, neuroprotective, hepatoprotective, antidiabetic, antifungal, antibacterial, and antiviral activities [7,12]. Their effects have been attributed to many components, such as minerals, essential amino acids, dietary fiber, proteins, polysaccharides, lipopolysaccharides, glycoproteins, and secondary metabolites. Among these, some of the complex organic compounds have shown immunomodulatory effects [7]. For example, polysaccharides from MMs can activate natural killer cells, macrophages, and neutrophils, as well as induce innate immune cytokines and interleukins. In addition, secondary metabolites from MMs, such as sterols, terpenes, and phenols can enhance the survival of hosts by stabilizing their important metabolic functions [8].

Different MMs contain different functional components that may impact the same or different immunomodulatory pathways at varying efficacies. In the following sections, we first provide a brief description of the known MMs. We then summarize the diversity, structure, function, and molecular mechanism of action of functional ingredients from MMs that have shown to be involved in immunomodulation. We finish by briefly describing how genomics can accelerate research on medicinal mushrooms.

## 2. Medicinal Mushrooms

As mentioned above, medicinal mushrooms refer to all macroscopic fungi whose extracts or powder form from any stages of the mushroom development have shown documented beneficial effects on health [13]. These beneficial effects may have been shown in the forms of in vitro, ex vivo, or in vivo activities. Their effects may cover different groups of organisms such as antagonistic effects against human pathogens and parasites, and/or beneficial effects for human and animal cell lines, or animal and human individuals [11]. Since many edible mushrooms and their products have shown to be a beneficial component of the human diet, some of these edible mushrooms are also commonly included as medicinal mushrooms [7]. In our literature search, a large number of MMs have been documented. For example, terpenes and terpenoids from *Ganoderma lucidum* could stimulate the expressions of genes coding for proteins in the nuclear factor (NF)–kB pathway and modulate immune system functions [14]. Heteroglycan and heteroglycan-peptide from the mushroom of *Hericium erinaceus* can modulate the immuno-effects by inducing nitric oxide production and increasing expression of tumor necrosis factor (TNF)-α, interleukin (IL)-1β, IL-12 [15]. These mushrooms belong to two fungal phyla, Ascomycota and Basidiomycota. Most of the MMs are in phylum Basidiomycota. Table 1 shows the major medicinal mushrooms, including their taxonomy and geographic/ecological distributions. As can be seen, some of these mushrooms are broadly distributed (e.g., the button mushroom *Agaricus bisporus*) while others are geographically more restricted (e.g., the Himalayan caterpillar fungus *Ophiocordyceps sinensis*). Some of the mushrooms included in the table e.g., *Amanita phalloides* are highly poisonous when consumed by humans. However, the dilutions of an *A. phalloides* extract that contains the toxin amanitin have shown to be effective as an anti-tumor therapy [16].

Some of these MM species have been used as herbal medicine for centuries, including *Ganoderma lucidum*, *Ganoderma lingzhi*, *Lentinula edodes*, *Inonotus obliquus*, *Fomitopsis officinalis*, *Piptoporus betulinus*, and *Fomes fomentarius* [7,17]. While these mushrooms have attracted most of the medical attention among the MMs, other species in multiple genera have also shown immunomodulatory and anti-tumor effects, such as those in genera *Agaricus*, *Albatrellus*, *Antrodia*, *Calvatia*, *Clitocybe*, *Cordyceps*, *Flammulina*, *Fomes*, *Funlia*, *Ganoderma*, *Inocybe*, *Inonotus*, *Lactarius*, *Phellinus*, *Pleurotus*, *Russula*, *Schizophyllum*, *Suillus*, *Trametes*, and *Xerocomus* [12].

Figure 1 shows a few representative medicinal mushroom species in their natural habitats. Some medicinal mushrooms are only found in the wild, e.g., the ectomycorrhizal mushrooms *Boletus edulis* and *Russula lepida*. However, a large number of medicinal mushrooms are also commercially cultivated, including Shiitake, Ling-zhi, and Lion’s Mane. Figure 2 shows a few representative medicinal mushrooms under cultivation.

## 3. Immunomodulatory Compounds and Commercial Products of Medicinal Mushrooms

As shown above, there is a large number of medicinal mushrooms in diverse taxonomic groups. Some of these medicinal mushrooms are commercially cultivated for food but there is an increasing trend for developing the immune-active compounds from these cultivated mushrooms into nutraceuticals. Table 2, Table 3, Table 4 and Table 5 summarize the major groups of bioactive compounds in medicinal mushrooms and their demonstrated immunomodulatory effects to specific pathologies, including the relevant references.

The main classes of compounds from medicinal mushrooms with immunomodulatory properties are terpenes and terpenoids, lectins, fungal immunomodulatory proteins (FIPs), and polysaccharides (particularly β-d-glucans, but also include polysaccharopeptides and polysaccharide proteins) [1]. Below we describe specific examples in each of these groups.

### 3.1. Polysaccharides

Among the bioactive compounds derived from mushrooms with immunomodulatory activity, those based on polysaccharides, with or without side chain modifications (including polysaccharopeptides and polysaccharide proteins) are the most reported during the last several decades [18]. Table 2 presents a list of polysaccharides from medicinal mushrooms that have shown immunomodulatory activities. Among the reported polysaccharides with immunomodulatory and antitumor activities, the best-known is lentinan, isolated from shiitake (*L. edodes*), as well as schizophyllan from *Schizophyllum commune*. Both lectinan and schizophyllan contain β-1,3-d-glucans with β-1,6 branches. Specifically, lentinan showed immunomodulatory properties against gastric cancer while schizophyllan was effective against head and neck cancer. Both products have been licensed and approved in Japan since 1986 for clinical use, in combination with chemotherapy against the two respective cancers [19,20].

Other polysaccharide-based compounds showing immunomodulatory properties have a similar core polysaccharide chemical structure but contain different branching linkages and/or branches with different conjugates. These polysaccharide-conjugate complexes are called heteroglucans, with α(1–4)- and β(1–3) glycosidic linkages to protein components. For example, in the presence of fucose (a hexose deoxy sugar with the chemical formula C6H12O5), the turkey tail mushroom *Trametes versicolor* produces a Krestin bound β-glucan polysaccharide K (PSK). PSK is commercially produced from this mushroom in Japan and has been approved for clinical use since 1977 [21]. Several subsequent reports confirmed the effectiveness of PSK as an adjuvant to conventional cancer therapies through inhibition of cancer metastasis [22], induction of cancer cell apoptosis [23], improvement of inflammatory cytokines gene expression [24,25].

Another compound isolated from *T. versicolor* is a polysaccharide peptide (PSP). PSP contains rhamnose and arabinose, two monosaccharides not found in PSK. In addition, the conjugated protein was also different. PSP has been commercially available in the Chinese market since 1987 [21]. It has been documented to improve the quality of life in cancer patients by providing substantial pain relief and enhancing immune status in 70–97% of patients with stomach, esophagus, lung, ovary and cervical cancers. Specifically, PSP has been shown to be capable of boosting immune cell production, ameliorating chemotherapy symptoms, and enhancing tumor infiltration by dendritic and cytotoxic T-cells [26].

Two other well-known polysaccharide–protein complexes produced by *Macrocybe gigantea* (syn. *Tricholoma giganteum*) and *Agaricus brazei*, respectively, have also shown immunomodulatory effects. A polysaccharide-protein complex (PSPC) isolated from *T. giganteum* showed that it could help restore and improve the phagocytic function of macrophages in tumor-bearing mice [13,27]. Similarly, AbM isolated from *A. brazei* contains diverse polysaccharide–protein complexes with different chemical linkages such as β-1,6-glucan, α-1,6- and α-1,4-glucan, glucomannan and β-1,3-glucan. AbM has been shown to have immunomodulatory and antineoplastic properties [28]. Polysaccharides and polysaccharide–protein complexes from other medicinal mushrooms that have shown immunomodulatory effects are listed in Table 2.

**Table 2 jof-06-00269-t002:** Major immunomodulatory polysaccharides from medicinal mushrooms and their immunomodulatory effects.

Source	Active Compound	Immunomodulatory Effect	Refs
*Agaricus blazei* (syn. *Agaricus brasiliensis*)	Heteroglycan, Glycoprotein, Glucomannan-protein complex, β-1,3-d-glucan, with β-1,6-d-glucan branch	Stimulates Natural Killer (NK) cells, macrophages, dendritic cells, and granulocytes; induction of Tumor Necrosis Factor (TNF), Interferon (IFN)-γ, and Interleukin (IL)-8 production	[29]
*Auricularia auricula-judae*	AF1 β-1,3-d-glucan main chain with two β-1,6-d-glucosyl residues	Induces apoptosis of cancer cell	[30]
*Gymnopus dryophilus* (syn. *Collybia dryophila*)	β-d-glucan	Inhibits NO production in activated macrophages	[31]
*Ophiocordyceps sinensis*	β-d-glucan, heteroglycan, cordyglucan	Increase in IL-5 induction with decrease in IL-4 and IL-17	[32]
*Cryptoporus volvatus*	β-1,3-d-Glucan	Decreases in TLR2 and activate NF-κB	[33]
*Flammulina velutipes*	Glycoprotein, *Flammulina velutipes* peptidoglycan (FVP), β-1,3-d-glucan	Increases NO, IL-1 production, and TNF-α secretion	[34]
*Ganoderma lucidum*	Ganoderan, Heteroglycan, mannoglucan, glycopeptide	Stimulates TNF-α, IL-1, IFN-γ production, activate NF-κB.	[35]
*Grifola frondosa*	Grifolan (1–6-monoglucosyl-branched β-1,3-d-glucan), proteoglycan, heteroglycan, galactomannan	Macrophage activation, induction of IL-1, IL-6, and TNF-α secretion	[36]
*Hericium erinaceus*	Heteroglycan, heteroglycan-peptide, β-1,3 branched-β-1,2-mannan	Induces NO production, increase expression of TNF-α, IL-1β, IL-12	[15]
*Inonotus obliquus*	β-d-glucan	Enhance expression of IL-1β, IL-6, TNF-α, and inducible nitric oxide synthase (iNOS) in macrophages	[37]
*Lentinula edodes* (syn. *Lentinus edodes*)	Lentinan, glucan, mannoglucan, proteoglycan, β-(1-6)-d-glucan, α-(1-3)-d-glucan	Induces non-specific cytotoxicity in macrophage and enhance cytokine production Induces cytotoxic effect on a breast cancer cell line	[38,39,40]
*Lentinus squarrosulus*	Glucan	Activation of macrophages, splenocytes and thymocytes	[41]
*Morchella esculenta*	Galactomannan, β-1,3-d-glucan	Macrophage activation, activate NF-κB	[42]
*Morchella conica*	Galactomannan	Induces NO, IL-1β, IL-6 production	[43]
*Naematelia aurantialba* (syn.* Tremella aurantialba)*	Heteroglycan	Enhances mouse spleen lymphocyte proliferation	[44]
*Pleurotus* sp. *‘Florida’* (syn.* Pleurotus florida*)	α-1,6-glucan and α-1, 3-, β-1,6-d-glucan	Stimulates macrophages, splenocytes and thymocytes	[45,46]
*Pleurotus ostreatus*	Pleuran, heterogalactan, proteoglycan	Induces IL-4 and IFN-γ production	[47]
*Poria cocos*	β-pachyman, β-Glucan, β-1,3-d-glucan, α-1, 3-d-glucan	Promotes the immune reaction; increases the expression of cytokines	[48,49]
*Sarcodon aspratus*	Fucogalactan, 1,6-α-d-glucopyranosyl residue	Increases the release of TNF-α and NO in macrophage	[50]
*Schizophyllum commune*	Schizophyllan, 1,6-monoglucosyl branched β-1, 3-d-glucan	Activation of T cell, increases interleukin, and TNF-α production	[51]
*Sparassis crispa*	β-Glucan	Enhances IL-6 and INF-γ production	[52]
*Taiwanofungus camphoratus* (syn. *Antrodia camphorate)*	β-1,3-d-Gluco-pyranans with β-1,6-d-glucosyl branches, proteoglycan	Induction of INF-γ, TNF-α	[53]
*Tropicoporus linteus (*syn.* Phellinus linteus*)	Acidic polysaccharides	Activation of murine B cells, Induces IL-12 and IFN-γ production, Blocks NF-κB, TNF-α, IL-1α, IL-1β, and IL-4 production	[54]
*Trametes versicolor*	Polysaccharide peptide Krestin (PSK), β-1,3-glycosidic bond with β-1,6-glycosidic branches	Increases the expression of cytokines; stimulates the macrophage phagocytes	[1,55]
*Tremella fuciformis*	Acidic glucuronoxylomannan α-1,3-d-mannan backbone with β-linked D-glucuronic acid	Induces human monocytes to express interleukins	[56,57]
*Macrocybe gigantea* (syn.* Tricholoma giganteum*)	Polysaccharide-protein complex (PSPC)	Increases phagocytic function of macrophages by activating macrophages to release mediators such as NO and TNF-α and inhibits S180 and HL-60 cells	[13,27]
*Xylaria nigripes*	β-Glucan	Inhibits NO, IL-1β, IL-6, TNF-α, and IFN-γ production	[58]

### 3.2. Mushroom Proteins and Protein–Conjugate Complexes

Mushroom proteins and protein–conjugate complexes are also well-known as immunomodulatory compounds. Similar to the polysaccharide-based compounds, these protein-based immunomodulatory compounds in medicinal mushrooms can also be grouped into different categories. Here, these compounds are grouped into two major categories: fungal immunomodulatory proteins (FIPs) and lectins. FIPs differ from lectins by having no conjugate while each lectin contains specific carbohydrates conjugated to a polypeptide.

Table 3 lists all the lectins from medicinal mushrooms isolated so far that have shown immunomodulatory effects. These lectins have been shown to be capable of stimulating nitrite production, upregulating the expressions of tumor necrosis factor (TNF)-α and interleukins, activating lymphocytes, and promoting the production of macrophage-activating factors etc. The medicinal mushroom species containing such lectins are very diverse, including *Floccularia luteovirens* (syn. *Armillaria luteovirens*), *Ganoderma capense*, *Grifola frondosa*, *Pseudosperma umbrinellum* (syn. *Inocybe umbrinella*), *Pholiota adipose*, *Pleurotus citrinopileatus*, *Russula delica*, *S. commune*, *Leucocalocybe mongolica* (syn. *Tricholoma mongolicum*), *Volvariella volvacea*, and *Xerocomus spadiceus* [59,60,61,62,63,64]. In addition, several mushroom lectins have also shown potent antiviral, mitogenic, antimicrobial and antioxidant activities [59,63,65,66,67,68,69].

Similarly, a large number FIPs have been identified. The FIP names, the medicinal mushrooms that produce them, and evidence for their specific immunomodulatory effects are presented in Table 4. Among these, the best known is probably Ling-Zhi-8 from *G. lucidum* which acts as an immunosuppressive agent [1]. In addition, aside from immunomodulation, many FIPs have also shown antitumor activities in pharmacological tests, including the inhibition of cell growth and proliferation, the induction of apoptosis and autophagy, and the reduction of invasion and migration of tumor cells. At present, most of these tests are conducted using tissue cultures. Further tests using animal models and clinical trials are needed in order to confirm their safety and efficacy in humans. If confirmed, these FIPs could be more efficiently produced and commercialized through genetic engineering for clinical use.

**Table 3 jof-06-00269-t003:** Major immunomodulatory lectins from medicinal mushrooms and their immunomodulatory effects.

Source	Lectin name	Immunomodulatory effect	Refs
*Agaricus bisporus*	*Agaricus bisporus* lectin (ABL)	Stimulate mice splenocytes mitogenicity and inhibit proliferation of L1210 and HT-29 cells	[70,71]
*Agrocybe aegerita*	*Agrocybe aegerita* lectin (AAL)	Inhibit proliferation of 4T1, HeLa, SW480 SGC7901, MGC803, BGC823, HL-60 and S180 cells	[72,73]
*Amanita phalloides*	-	Inhibit proliferation of L1210 cells	[74]
*Floccularia luteovirens (syn. Armillaria luteovirens)*	*Armillaria luteovirens* lectin (ALL)	Stimulate mice splenocytes mitogenicity and inhibit proliferation of L1210, Mouse myeloma MBL2 and HeLa cells	[75]
*Boletus edulis*	*Boletus edulis* lectin (BEL)	Stimulate mice splenocytes mitogenicity and inhibit proliferation of human hepatocyte carcinoma G2 (HepG2) and HT-29 cells	[76]
*Boletus speciosus*	*Boletus speciosus* hemagglutinin (BSH)	Inhibit proliferation of HepG2 and L1210 cells	[77]
*Clitocybe nebularis*	*Clitocybe nebularis* lectin (CNL)	Inhibit proliferation of human leukemic T cells	[78]
*Flammulina velutipes*	*Flammulina velutipes* agglutinin (FVA)	Stimulate mice splenocytes mitogenicity and inhibit proliferation of L1210 cells	[79]
*Ganoderma capense*	-	Stimulate mice splenocytes mitogenicity and inhibit proliferation of L1210, M1, HepG2 cells	[62]
*Grifola frondosa*	*Grifola frondosa* lectin (GFL)	Inhibit proliferation of HeLa	[80]
*Hericium erinaceus* (Syn. *Hericium erinaceum*)	*Hericium erinaceus* agglutinin (HEA)	Inhibit proliferation of HepG2 and human breast cancer MCF7 cells	[81]
*Kurokawa leucomelas*	*Kurokawa leucomelas* KL-15	Inhibit proliferation of U937 cells	[82]
*Lactarius flavidulus*	*Lactarius flavidulus* lectin (LFL)	Inhibit proliferation of HepG2 and L1210 cells	[83]
*Lignosus rhinocerotis*	*Lignosus rhinocerotis* lectin (LRL)	Inhibit proliferation of HeLa, MCF7 and A549 cells	[84]
*Marasmius oreades*	*Marasmius oreades* agglutinin (MOA)	Inhibit proliferation of SW480, HepG2 and NIH-3T3 cells	[85]
*Pholiota adiposa*	*Pholiota adiposa* lectin (PAL)	Inhibit proliferation of HepG2 and MCF7 cells	[61]
*Pleurotus citrinopileatus*	-	Stimulate mice splenocytes mitogenicity and inhibit proliferation of S180 cells	[59]
*Pleurotus eous*	*Pleurotus eous* lectin (PEL)	Inhibit proliferation of MCF7, K562 and HepG2	[86]
*Cerioporus squamosus* (syn. *Polyporus squamosus*)	*Polyporus squamosus* lectin 1a (PSL1a)	Inhibit proliferation of HeLa cells	[87]
*Pseudosperma umbrinellum* (syn. *Inocybe umbrinella*)	*Inocybe umbrinella* lectin (IUL)	Inhibit proliferation of HepG2 and MCF7 cells	[60]
*Russula delica*	-	Inhibit proliferation of HepG2 and MCF7 cells	[64]
*Russula lepida*	*Russula lepida* lectin (RLL)	Inhibit proliferation of HepG2 and MCF7 cells	[88]
*Schizophyllum commune*	*Schizophyllum commune* lectin (SCL)	Stimulate mice splenocytes mitogenicity and inhibit proliferation of KB, HepG2 and S180 cells	[63,89]
*Stropharia rugosoannulata*	*Stropharia rugosoannulata* lectin (SRL)	Inhibit proliferation of HepG2 and L1210 cells	[90]
*Leucocalocybe mongolica* (syn.* Tricholoma mongolicum*)	*Tricholoma mongolicum* lectin 1 (TML-1), *Tricholoma mongolicum* lectin 2 (TML-2)	Inhibit proliferation of S180 cells	[91]
*Volvariella volvacea*	*Volvariella volvacea* lectin (VVL)	Inhibit proliferation of S180 cells and enhance IL-2 and IFN-γ transcriptions	[92,93]
*Xerocomellus chrysenteron* (syn. *Xerocomus chrysenteron*)	*Xerocomus chrysenteron* lectin (XCL)	Inhibit proliferation of NIH-3T3 and HeLa cells	[94]
*Xylaria hypoxylon*	*Xylaria hypoxylon* lectin (XHL)	Inhibit proliferation of HepG2 cells	[95]

**Table 4 jof-06-00269-t004:** Major fungal immunomodulatory proteins (FIPs) from medicinal mushrooms and their immunomodulatory effects.

FIP Name	Source	Immunomodulatory Effect	Refs
FIP-aca	*Taiwanofungus camphoratus* (Syn.* Antrodia camphorate*)	Induce expression of different cytokines (IL-1b, IL-6, IL-12, TNF-α) and chemokines (CCL3, CCL4, CCL5, CCL10)	[96]
FIP-cru1	*Chroogomphis rutilus*	Stimulate the proliferation of murine splenocytes and enhanced the secretion of IL-2	[97]
FIP-dsq (FIP-dsq2)	*Dichomitus squalens*	Induce apoptosis and interrupt migration of A549 cells	[98]
FIP-fve	*Flammulina velutipes*	Stimulate mitogenesis in human peripheral lymphocytes, suppress systemic anaphylaxis reaction, enhance transcription of IL-3, INF-g	[99]
FIP-gja	*Ganoderma japonicum*	-	GenBank: AAX98241
FIP-gat	*Ganoderma atrum*	-	[100]
FIP-glu1 (LZ-8)	*Ganoderma lucidum*	Enhance transcription of IL-2, IL-3, IL-4, IFN-g, TNF-α	[101]
FIP-gmi	*Ganoderma microsporum*	Down regulation of TNF-α	[102]
FIP-gsi	*Ganoderma sinensis*	Enhance production of IL-2, IL-3, IL-4, INF-g, TNF-a	[103]
FIP-gts	*Ganoderma tsugae*	Induce cytokine secretion, cellular proliferation of human peripheral mononuclear cells (HPBMCs), enhance IFN-g expression	[104]
FIP-glu2 (LZ-9)	*Ganoderma lucidum*	Activate THP-1 macrophages and induce pro-inflammatory cytokine transcription	[105]
FIP-SN15	Intrageneric shuffled library	Induce U-251 MG cells apoptosis	[106]
FIP-Irh	*Lignosus rhinocerotis*	Inhibit the proliferation of MCF7, HeLa and A549 cancer cell lines	[84]
FIP-pcp	*Poria cocos*	Enhance production of IL-1b, IL-6, IL-18, TNF-a, NO	[107]
FIP-ppl	*Postia placenta*	Stimulate mouse splenocyte cell proliferation and enhance interleukin-2 (IL-2) release, inhibit proliferation and induce apoptotic effects on gastric tumor cells (MGC823)	[108]
FIP-tve2 (FIP-tvc)	*Trametes versicolor*	Increase human peripheral blood lymphocytes, enhanced production of TNF-a, NO	[109]
FIP-vvo	*Volvariella volvacea*	Enhance expression of IL-2, IL-4, IFN-g, TNF-a	[110]

“-” not tested.

### 3.3. Terpenes and Terpenoids

Terpenes are a large and diverse class of hydrocarbon compounds derived biosynthetically from units of isopentenyl pyrophosphate. They are widespread in nature, produced by a variety of plants, particularly conifers, some insects, and fungi, including mushrooms. The addition of functional groups (usually oxygen-containing) to terpenes produce terpenoids. Both terpenes and terpenoids from a number of medicinal mushrooms have shown immunoregulatory activities with medical significance. Table 5 shows the types of terpenes and terpenoids that have been isolated from medicinal mushrooms, including evidence for their specific immunomodulatory activities. For example, *Ganoderma* sp. are known for their high content of triterpenoids and these triterpenoids have shown high immunomodulating and anti-infective activities [111,112,113]. A study showed that terpenes and terpenoids modulate immune system functions by stimulating the expressions of genes coding for proteins in the nuclear factor (NF)–kB pathway and for mitogen-activated protein kinases [14].

**Table 5 jof-06-00269-t005:** Major immunomodulatory terpenes and terpenoids from medicinal mushrooms and their immunomodulatory effects.

Type of Terpenes	Source	Compound	Immunomodulatory Effect	Refs
Monoterpenoids	*Pleurotus cornucopiae*	-	Inhibit the proliferation of HeLa and HepG2 cells	[114]
Sesquiterpenoids	*Stereum hirsutum*	-	Inhibit the proliferation of HepG2 and A549 cells	[111]
	*Inonotus rickii*	3α,6α-Hydroxycinnamolide	Inhibit the proliferation of SW480 cells	[112]
	*Pleurotus cornucopiae*	Pleurospiroketals A, B, C	Inhibit the proliferation of HeLa cells	[113]
	*Anthracophyllum* sp. *BCC18695*	Anthracophyllone	Inhibit the proliferation of MCF7, NCI-H187, KB and Vero cells	[115]
	*Flammulina velutipes*	Enokipodins B, D, J 2,5-cuparadiene-1,4-dione Flammulinolides A, B, C, F	Inhibit the proliferation of HepG2, MCF7, SGC7901, KB, HeLa and A549 cells	[116,117]
	*Neonothopanus nambi*	Nambinones C	Inhibit the proliferation of NCI-H187 cells	[118]
Diterpenoids	*Cyathus africanus*	Neosarcodonin O, 11-O-acetylcyathatriol, Cyathins H	Inhibit the proliferation of K562 and Hela cells	[119]
	*Pleurotus eryngii*	Eryngiolide A	Inhibit the proliferation of Hela and HepG2 cells	[120]
	*Sarcodon scabrosus*	Sarcodonin G	Inhibit the proliferation of HOC-21, HEC-1, U251-SP, MM-1CB and HMV-1 cells	[121]
	*Tricholoma* sp.	Tricholomalide A, B, C	Inhibit the proliferation of HeLa cells	[122]
Triterpenoids	*Ganoderma boninense*	Ganoboninketals A, B, C	Inhibit the proliferation of A549 and HeLa cells	[123]
	*Ganoderma orbiforme BCC 22324*	Ganoderic acid T and its C-3 epimer compound	Inhibit the proliferation of NCIH187, MCF7 and KB cells	[124]
	*Ganoderma lucidum*	lucialdehydes B, C, ganodermanondiol, ganodermanonol, ganoderic acid DM, ganoderic acid X	Inhibit the proliferation of T-47D, LLC, Meth-A, and Sarcoma 180 cells; Decrease the protein levels of CDK2, CDK6, p-Rb, cycle D1 and c-Myc in MCF7 cells; inhibit activity against topoisomerases I and II α and promote apoptosis	[125,126,127]
	*Ganoderma concinna*	5α-lanosta-7,9(11),24-triene-3β-hydroxy-26-al, 5α-lanosta-7,9(11),24-triene-15α-26-dihydroxy-3-one, 8α,9α-epoxy-4,4,14α-trimethyl-3,7,11,15,20-pentaoxo-5α-pregrane	Induce apoptosis in promyelocyticleukemia HL-60 cells	[128]
	*Ganoderma tsugae*	Tsugaric acid A, 3β-hydroxy-5α-lanosta-8, 24-dien-21-oic acid	Inhibit the proliferation of HT-3, T-24, and CaSKi cells	[129]
	*Hypholoma fasciculare* (syn. *Naematoloma fasciculare*)	Fusciculol C, L, M, G	Inhibit the proliferation of HCT-15, SK-OV-3, SK-MEL-2 and A549 cells	[130]
	*Astraeus odoratus*	Astraodoric acids A, B, D	Inhibit the proliferation of KB, NCI-H187, and MCF7 cells	[131]
	*Russula lepida Russula amarissima*	Cucurbitane hydroxyl acid	Inhibit the proliferation of WISH, CAKI 1 and A549 cells	[132]
	*Leucopaxillus gentianeus*	Cucurbitacin B Leucopaxillone A	Inhibit the proliferation of MCF7, HepG2, kidney carcinoma CAKI-1 and A549 cells	[133]
	*Hebeloma versipelle*	24(E)-3β-hydroxylanosta-8,24-dien-26-al-21-oic acid	Inhibit the proliferation of HL60, Bel-7402, SGC-7901 and A549 cells	[134]
	*Tricholoma saponaceum*	Saponaceol A	Inhibit the proliferation of HL-60 cells	[135]
	*Elfvingia applanata*	The methyl ester of elfvingic acid H	Inhibit the proliferation of Ehlrich and Kato III cells	[136]

## 4. Immunomodulation and Other Human Health Effects of Medicinal Mushrooms

The human immune system is tightly linked to tumor development. With the increasing impact of tumor on human health, a large number of studies have been undertaken to identify mushroom extracts/fractions/compounds with antitumor activities. Indeed, some of the observed antitumor activities by medicinal mushroom extracts were based on the activation of the immune system (Table 2, Table 3, Table 4 and Table 5). Davis et al. (2020) recently suggested that 17 medicinal mushroom species (*A. brazei*, *Cordyceps militaris*, *Flammulina velutipe*, *F. fomentarius*, *F. officinalis*, *Ganoderma applanatum*, *G. lucidum*, *Ganoderma oregonense*, *G. frondosa*, *Hericium erinaceus* (syn. *Hericium erinaceum*), *I. obliquus*, *L. edodes*, *Tropicoporus linteus* (syn. *Phellinus linteus*), *P. betulinus*, *Pleurotus ostreatus*, *S. commune*) could support both immune-activation for cancer treatments and help resolve host defense-induced inflammatory reactions and facilitate a post-response return to homeostasis for cancer patients. Furthermore, a medicinal mushroom formulation consisting of *G. lucidum*, *L. edodes* and *G. frondosa* showed synergistic antitumor and immuno-modulatory activity in human macrophages [137].

As shown in Table 2, Table 3, Table 4 and Table 5, many medicinal mushrooms each can produce different categories of compounds with immunomodulatory effects. Furthermore, different extractions of the same mushroom may show non-overlapping but complementary activities. For example, in *L. edodes*, its heterogalactan (fucomannogalactan) has anti-inflammatory properties [138], lentinan has anti-tumor effect [139], crude water-soluble polysaccharides can activate macrophages and increase the productions of nitric oxide (NO), cytokines, and proteins related to phagocytosis [140], and polysaccharides with both antioxidant effects [141] and antiviral activities [142].

The significance of functional components from medicinal mushrooms has been shown not only from clinical perspectives but also from foods. Because many medicinal mushrooms are commercially cultivated for food, there has been an increasing trend of including mushrooms and their components into other foods to develop functional foods, including adding new flavors or promoting certain types of functions. For example, Ulziijargal et al. used mushroom mycelia of *T. camphoratus*, *A. blazei*, *H. erinaceus*, and *P. linteus* to substitute 5% of wheat flour to make bread. The final product contains substantial amounts of the amino acids gaminobutyric acid (GABA) and ergothioneine and showed beneficial health effects [143]. Kim et al. developed noodles that contained *L. edodes* paste, resulting in a higher quality, fibre-rich functional food with antioxidant and hypocholesterolemic properties [144]. Components of other mushrooms, e.g., *Pleurotus sajor-caju* dry powder, *A. bisporus* extracts, the freeze-dried powder from *A. aegerita*, *Suillus luteus*, and *Coprinopsis atramentaria* have been used to develop snacks and cheese-related products with much commercial success [145,146,147,148]. The benefits include increased proteins, minerals, crude fiber or ingredients with antioxidant potentials and free radical scavenging capacities. The diversified flavors and tastes as well as the enhanced nutritional values of food due to the addition of mushroom components represent an exciting direction for the edible and medicinal mushroom industries.

## 5. Mechanisms for the Immunomodulation Effects of Medicinal Mushroom Compounds

The immune system consists of a network of cells, tissues and organs that work together to defend the body against attacks by “foreign” invaders [2]. The network is connected by lymphatic vessels from organ to organ. The network includes protective barriers that constantly communicate with lymphatic fluid rich in white blood cells and leukocytes. When pathogens break our physical barriers (i.e., skin and mucosal membranes of the mouth, nose, the gastrointestinal tract, and the urogenital tract), the next line of the body’s defense response is activated. This line of defense includes granulocytes and monocytes that also function as antigen-presenting cells (APCs) for helper T lymphocytes. These cells synthetize and secrete lipid mediators such as prostaglandins as well as cytokines which act as messengers in regulating immune response and stimulating adaptive immunity. For example, natural killer (NK) cells can recognize infected and abnormal cells, such as cancer cells and kill these cells by inducing them to undergo apoptosis or by producing cytokines, such as interferon-gamma (IFN-γ). They also activate macrophages and kill phagocytosed microbes. Figure 3 presents the overview of the key immune response against microbial pathogens.

It is well known that the human immune system can be modulated by foods, supplements or endogenous bioactive agents [2]. Different types of immunoregulatory compounds have been isolated from medicinal mushrooms, including mushroom fruiting bodies and fermented mycelia. Table 2, Table 3, Table 4 and Table 5 present the specific components isolated from different mushroom species that have shown significant immunomodulatory activities, including their (potential) mechanisms of action. For example, two polysaccharides from two different mushroom species have shown significant immunoenhancing activities. In the first, a glucuronoxylomannan TAP-3 obtained from *Naematelia aurantialba* (syn. *Tremella aurantialba*) showed marked immune enhancement activity and promoted NO, IL-1β and TNF-α secretions from macrophages [149]. Similarly, another study showed that at a concentration of 40 μg/mL, polysaccharide CCP from *Craterellus cornucopioides* strengthened the phagocytic function of macrophages, increased the expression of cytokines, upregulated the expression of cell membrane receptor TLR4 and downstream protein kinase products through activation of the TLR4–NFκB pathway [150].

Some of these bioactive compounds can also directly attack cancer cells while showing immunoregulatory effects. For example, Li et al. reported that polysaccharide LRP-1 purified from *Leccinum rugosiceps* inhibited the growth of human hepatoma cells HepG2 and human breast carcinoma MCF-7 cells, and induced the secretion of NO, IL-6 and TNF-α in vitro [151]. Similarly, a recent report showed that an aqueous extract of *Sarcodon imbricatus* (SIE) effectively inhibited the growth, migration, and invasion properties of breast cancer cells in vitro and reduced tumor growth in vivo, while showing increased expression of PD-L1 and increased NK cell viability [152]. Furthermore, Xue et al. reported that a triterpenoid EAe from *Pleurotus eryngii* inhibited MCF-7 cell lines proliferation with an EC50 of 298.26 μg/mL, and significantly inhibited the growth of CD-1 tumors (inhibition rate of 65%) in mice in a dose-dependent manner without toxicity [153].

## 6. Relationship between Structure and Activity of Immunomodulatory Compounds from Medicinal Mushrooms

Immunomodulators from medicinal mushrooms have been shown to be capable of stimulating both innate and adaptive immune responses. They activate innate immune system components such as natural killer (NK) cells, neutrophils, and macrophages, and stimulate the expression and secretion of cytokines. These cytokines in turn activate adaptive immunity by promoting B cell proliferation and differentiation for antibody production and by stimulating T cell differentiation to T helper (Th) 1 and Th2 cells, which mediate cellular and humoral immunities, respectively [154].

As shown in Table 2, Table 3, Table 4 and Table 5, a large number of immunoregulatory compounds from medicinal mushrooms have been reported. These compounds differ greatly in their molecular weight and structure. Below we describe the relationships between their molecular structures and immunoregulatory activities.

### 6.1. Polysaccharides

Polysaccharides are the most commonly reported natural immunomodulators from mushrooms. The immunomodulating polysaccharides are highly diversified in their sugar compositions, main chain polymer structures, degrees of branching, conformations, molecular weights, and other physical properties, which together have significant effects on the bioactivity and mode of action of each polysaccharide [18]. These polysaccharides are either homoglycans (polysaccharides that contain residues of only one type of monosaccharide molecule) or heteroglycans (polysaccharides that contain residues of two or more types of monosaccharide molecules), and are able to combine with other molecules such as oligo- or poly-peptides to make peptidoglycan or polysaccharide–protein complexes. In general, higher molecular weight polysaccharides exhibit greater bioactivity [155]. These large polysaccharides are not able to penetrate the immune cells, but instead act to bind cell receptors. For example, the highest immunomodulatory activity of PSK was associated with the highest molecular weight fraction of this compound, at >200 kDa [156]. Similarly, the highest activity of a polysaccharide fraction of *G. frondosa* extract was ascribed to one with a molecule weight of over 800 kDa [157]. In contrast, low molecular weight polysaccharides can penetrate immune cells and exert stimulatory effects from within.

The number and lengths of short branched chains in mushroom polysaccharides can significantly influence their bioactivity [155]. In most cases, the bioactive immunomodulator polysaccharides are characterized by a main chain of 1,3-β-d-glucan with a small number of short branched chains with 1,6-β-linkage (Figure 4). Studies have shown that immunologically active polysaccharides generally have a degree of branching number (DB) between 20% and 40%. For example, lentinan has a DB number of 40%, schizophyllan of 33%, and PSK of 20%. While a high DB number is generally correlated with a high activity, in some cases, debranching of polysaccharides can also increase their bioactivity. For example, the partially debranched form of pachymaran from *Poria cocos* showed greater activity than the original native form [158]. Even in the well-studied lentinan, its maximal immunomodulating and antitumor activities were achieved when the molecule had a DB of 32% [159], and there was a negative correlation between their biological activity and DB number between 32 and 40% [160].

Aside from the main-chain structure and branching pattern, the conformation of polysaccharides can also impact their bioactivity, e.g., by influencing the stability of the structure. In polysaccharides, the triple helix conformation is usually more stable than other conformations and bears the cytokine stimulating activity of the β-d-glucan. Lentinan, schizophyllan, scleroglucan, and PSK all have a triple helix structure [161]. However, not all polysaccharide immunomodulators from mushrooms have the triple helix structure. For example, mushroom polysaccharides with a random coil conformation can also have potent immunomodulating and anticancer activity [56].

Chemical modification is an effective and common approach to increase biological activities of polysaccharides. This approach has been applied to develop a number of effective immunomodulators from mushroom polysaccharides. Those modifications include carboxymethylation, hydroxylation, formyl-methylation, amino-ethylation, or sulfation. The introduction of such chemical groups may increase the possible contacts between the modified polysaccharides and the immune cell receptors through hydrogen bonding and/or electrostatic attraction, and thus increase the immunological response. For example, the sulfated cell wall glucan from *L. edodes* exhibited higher immunomodulatory and anticancer activities compared to the native polysaccharides [162]. The increased effect from sulfation may be related to the increased solubility, as shown in the hyper branched β-glucan TM3b. Taken together, molecular weight, branching, chemical configuration, and chemical modification can all have strong influence on the bioactivity of polysaccharides from mushrooms.

### 6.2. Lectins

Lectins belong to a unique group of proteins that can recognize and interact with various cell surface carbohydrates/glycoproteins. Mushroom lectins have shown specific immunomodulatory, antiproliferative, and antitumor activities [163]. The diverse sources of mushroom lectins have different immunomodulatory mechanisms: some mediate their actions by activating the immune system while others produce potent cytotoxic effects towards cells [91]. For example, two lectins extracted from *L. mongolica* (syn.*T. mongolicum*), TML-1 and TML-2, show immunomodulatory and antitumor activities. These two lectins stimulate the production of nitrite and tumor necrosis factor (TNF)-α and inhibit the growth of mouse lymphoblast-like (p815) mastocytoma cells by the production of macrophage-activating factors. These factors include interferon (IFN)-γ and other cytokines, activated through upregulation of inducible nitric oxide synthase (NOS), interleukin (IL)-1β, and transforming growth factor-β [91]. *G. frondosa* lectin is reported as having a potent cytotoxic effect against HeLa cells in vitro, even at very low concentrations. A 15.9-kDa homodimeric, lactose-binding, ricin-B-like lectin (CNL) from *Clitocybe nebularis* exhibited antiproliferative activity against human leukemic T cells [78], which induces the maturation and activation of dendritic cells (DCs) and stimulates several proinflammatory cytokines such as IL-6, IL-8, and TNF-α [164]. The encoding gene of CNL from *C. nebularis* has been cloned and successfully expressed in *Escherichia coli* [165].

### 6.3. FIPs

The fungal immunomodulatory proteins are a group of proteins with highly similar amino acid sequences. They exist as dimers in a dumbbell-shaped structure similar to that of the variable region of immunoglobulin heavy chains [166]. The FIPs have shown diverse functions. Through binding to Toll-like receptors (TLRs), FIPs stimulate antigen presenting cells and release cytokines such as NO and IL-12. By activating phosphorylation of p38/MAPK and increasing the production of NF-κB, FIPs can promote the proliferation and differentiation of helper T cells (Th0) to form Th1 cells and Th2 cells, activate macrophages and B cells, produce a variety of cell factors (Figure 5). For example, FIP-fve from *Flammulina velutipes* can upregulate the expression of intercellular adhesion molecules on the T cell surface by phosphorylation of p38/MAPK, and activate Th1 cells to produce IL-2, IFN-γ, to exert its immunomodulatory effect [99]. FIP-vvo can not only activate Th1 cells and enhance IL-2, TNF-α and IFN-γ transformations, but also induce Th2 cells to produce IL-4, B cell differentiation, and the transformation of immunoglobulin and production of antibody IgE. Several studies have also shown that by interacting with TLRs, FIP can activate other signaling pathways besides the p38/MAPK and NF-κ B. For example, FIP from *Ganoderma tsugae* (FIP-gts) can stimulate human peripheral blood monocyte to produce IFN-γ and activates the PI3K/Akt signaling pathway [104].

FIPs typically exist in low quantities in their native mushrooms. The low yield/production has been a major limitation of their research and application. Therefore, techniques are being rapidly developed to enhance the production of recombinant FIPs in other organisms such as the yeast *Pichia pastoris* and the bacterium *E. coli*. For example, the LZ-8 gene of *G. lucidum* has been expressed in *P. pastoris* to produce a recombinant LZ-8 protein (rLZ-8). While the recombinant protein lacks the carbohydrate moiety of the native protein, it shows similar bioactivity for IL-2 induction as the native protein. The FIP-fve protein has also been successfully expressed in *E. coli* [1]. Interestingly, the recombinant FIPs showed higher immunomodulatory activity and induced greater expression of specific cytokines than that extracted from the mushrooms [167].

### 6.4. Terpenes and Terpenoids

Terpenes and terpenoids are widely distributed in mushrooms. They are a large and diversified group of organic compounds but share the core of isoprene five-carbon atom units of molecular formula (C_5_H_6_)n as the main building block (Figure 6) [1,13]. Among this group of compounds, the best-known is probably the triterpenoids from *G. lucidum* and *G. lingzhi*. These triterpenoids can help reduce drug nephrotoxicity and minimize inflammation. Figure 6 shows a diversity of terpene derivatives in *G. lucidum* and *G. lingzhi*, including ganodermic and ganoderic acids, ganoderals, ganoderols, ganodermanontriol, lucidone, and ganodermanondiol. All these compounds have shown immunomodulating, antitumor, and/or anti-infective activities [168]. At present, their mechanisms of action and structure–activity relationships are little understood. However, their broad activities suggest significant potential for research and for clinical therapeutic applications.

## 7. Genomes and Molecular Techniques in the Study of Immunomodulatory Compounds in Medicinal Mushrooms

Due to their environmental, agricultural, commercial, and/or medical interests, the genomes of a number of medicinal mushrooms have been sequenced and annotated. Table 6 lists the genomic features of 12 representative medicinal mushrooms. These species differ in genome size and/or gene content. Analyses of these genomes have revealed some of the genes related to the synthesis and production of immunomodulatory compounds in medicinal mushrooms. Not surprisingly, the most commonly identified genes related to immunomodulatory effects are those coding for FIPs (Table 6). However, in *G. lucidum*, genes involved in the synthesis of several other immunomodulators have also been identified. The identification and confirmation of those genes require genetic manipulation systems which are not available at present for most medicinal mushrooms. In *G. lucidum*, such a system is available.

Ganoderic acids (GAs) are among the main active ingredients of *G. lucidum* with immunomodulatory effects. GAs belong to the triterpenoid secondary metabolites. Genome sequence analyses and functional studies showed that the terpenoids in *G. lucidum* are synthesized through the Mevalonate (MVA) pathway. Several genes in this pathway in *G. lucidum* have been cloned and their functional roles confirmed, including those encoding 3-Hydroxy-3-methylglutaryl-CoA reductase (HMGR) and Farnesyl diphosphate synthase (FPPs). Genome sequence data mining also identified the putative genes involved in the modification of the triterpene backbone, such as those involved in cyclization and glycosylation, which are very important for the synthesis of the diversity of GAs in *G. lucidum*.

At present, most of the genes and metabolic pathways involved in the synthesis of immunomodulators in the categories of polysaccharides, lectins, and terpenoids in medicinal mushrooms have not been identified or confirmed. However, the availability of increasing genomic resources coupled with the broad pharmacological activities and therapeutic effects of medicinal mushrooms should help facilitate the identification of genes and metabolic pathways involved in their biosynthesis. Such understandings could help future productions of those compounds through biotechnology using surrogate hosts.

## 8. Conclusions and Perspectives

Many edible and medicinal mushrooms contain compounds with significant immunoregulatory activities. This paper attempts to provide a comprehensive review on the types of these compounds; their distributions, structures and functions; and their potential mechanisms of actions. These compounds have shown their activities through in vitro, ex vivo, tissue cultures, and/or in vivo studies. Some of these compounds have been commercialized and licensed for clinical use.

Aside from the above described compounds, other compounds from edible and medicinal mushrooms may also have great potentials. One such group is chitin and chitin-related compounds. Chitin is the most common aminopolysaccharide polymer in nature and the main material that gives strength to the fungal cell walls (as well as to the exoskeletons of crustaceans and insects). Through deacetylation, either chemically or enzymatically, chitin can be converted to chitosan, a well-known derivative. Through hydrolysis, both chitin and chitosan can be converted to chito-oligosaccharides. Several fungal chitin, chitosan, and chito-oligosaccharides have shown promising benefits to humans and human health [195,196]. For example, chitins from filamentous molds such as *Aspergillus niger* and *Mucor rouxii* and other organisms have been used in plant protection and food processing; chitosan in diagnosis, drug delivery, infection control, molecular imaging, and wound healing; and chito-oligosaccharides in antimicrobial and antitumor activities [195,196]. At present, none of those tested chitin, chitosan, and chito-oligosaccharides for human effects have come from edible or medicinal mushrooms yet. However, due to the similar chemical structures of chitin from different groups of organisms, it’s highly likely that this group of natural products from edible and medicinal mushrooms will have similar effects and they represent a promising area of future development for edible and medicinal mushrooms.

While the future looks bright, significant issues remain before the full potential of medicinal mushrooms can be reached. Specifically, during our review of the literature, we identified several significant gaps and areas for future research and development. In the first, there is an urgent need to identify the structures and mechanisms of action for active ingredients in many extracts and formulations from medicinal mushrooms. More rigorous chemical analyses as well as understanding the in vivo pharmacokinetics and pharmacodynamics of individual compounds are needed to fill this gap of knowledge. The second promising area of study is to identify the genes and metabolic pathways involved in producing these immunomodulators in medicinal mushrooms. As shown above, aside from the few FIPs where the specific encoding genes have been identified, we have little information about the genes and how they are regulated in producing the other types of mushroom immunomodulators. While the availability of high through-put technologies and genome sequences are facilitating the discoveries, experimental investigations are needed in order to confirm and identify the conditions for increased productions of these compounds. Fortunately, gene editing technologies and -omics tools are becoming increasingly accessible to the broader life sciences communities. In the third, most immunomodulators described above exist in low quantities in medicinal mushrooms and their extractions can take a long time and be costly. For efficient production, it is very important to develop alternative approaches, e.g., by cloning and expressing the relevant biosynthesis genes in alternative hosts, using industrial fermentations, and developing efficient extraction and purification protocols from such commercial cell cultures. Lastly, the potential interactions between immunomodulators from mushrooms and other medicines, foods, and food supplements need to be critically analyzed in order to establish guidelines for safe and effective use of immunomodulators from medicinal mushrooms [197]. Indeed, at present, safety data about many medicinal mushroom products are not available from controlled clinical trials and associated negative side effects have been reported in several cases for certain types of usages of medicinal mushroom products [198,199]. There is also a cultural difference between Oriental and Western cultures about the use of medicinal mushrooms [197], presenting both a challenge and an opportunity for researchers and policy makers on the broad implications of these products on human health.

## Figures and Tables

**Figure 1 jof-06-00269-f001:**
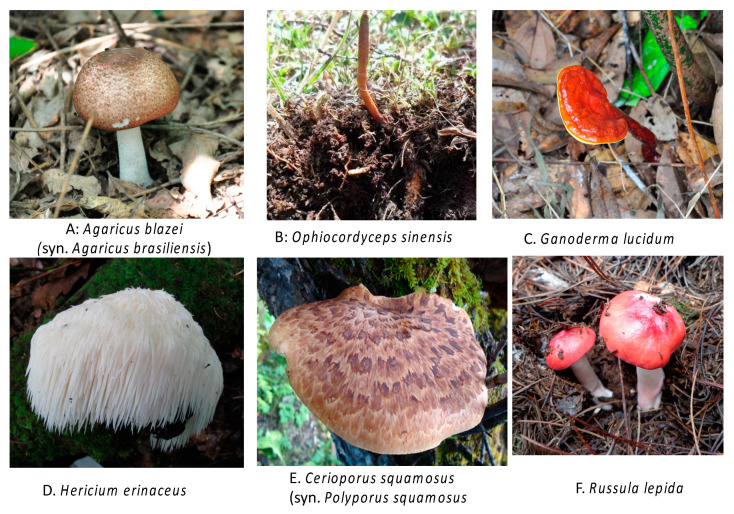
A few representative medicinal mushrooms from the wild.

**Figure 2 jof-06-00269-f002:**
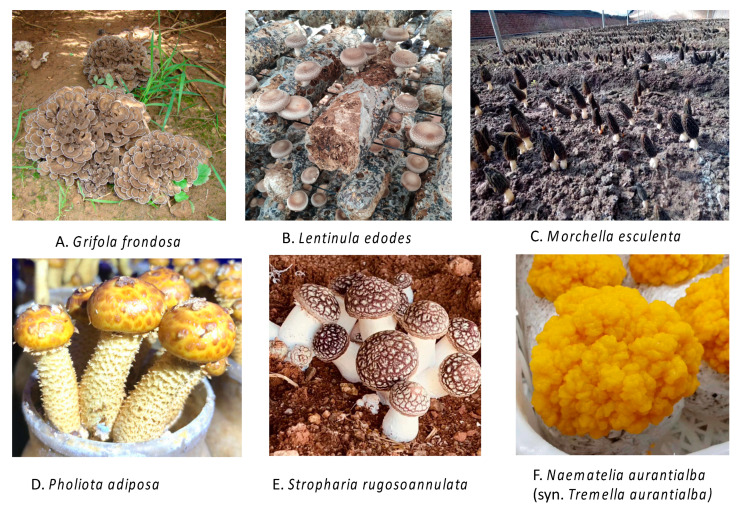
A few representative cultivated medicinal mushrooms.

**Figure 3 jof-06-00269-f003:**
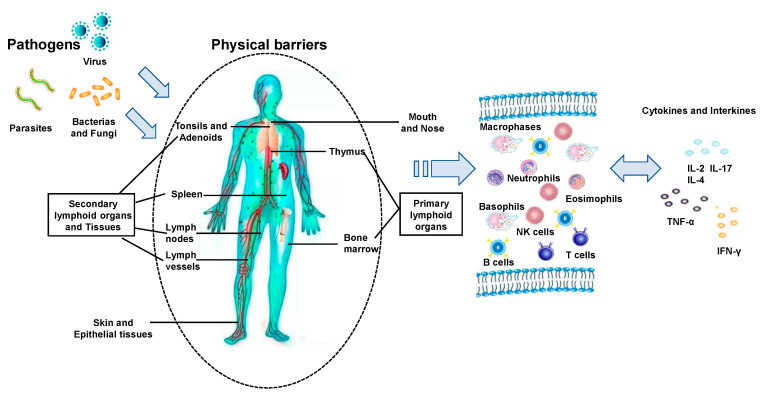
Key components and pathways in host immune response against pathogen infections.

**Figure 4 jof-06-00269-f004:**
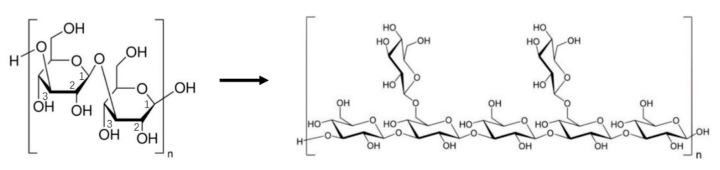
The basic structures of bioactive polysaccharides 1-3,-β-d-glucan and short branched chains with 1-6,-β-linkage.

**Figure 5 jof-06-00269-f005:**
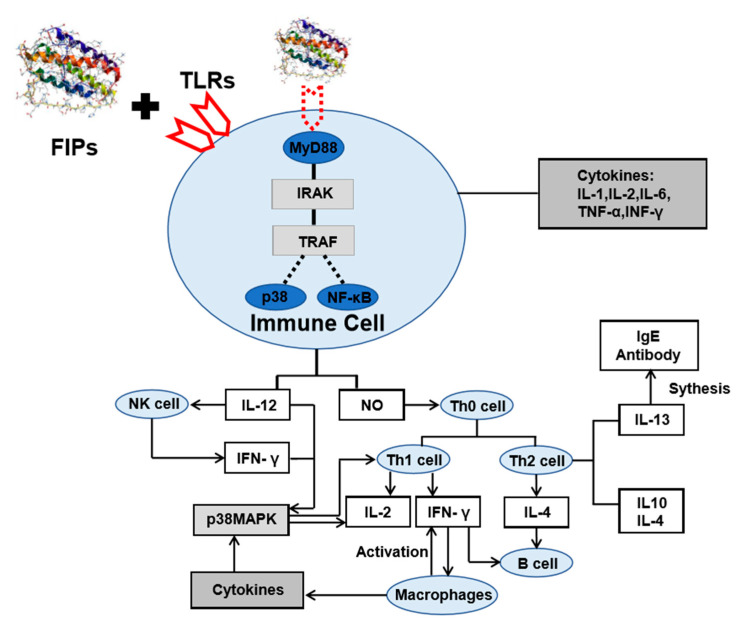
FIPs immunomodulatory mechanism by toll-like receptors (TLRs) signaling pathway.

**Figure 6 jof-06-00269-f006:**
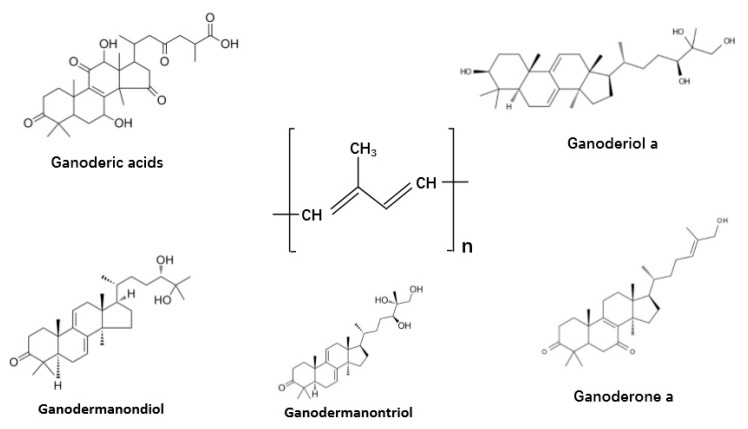
The structures of representative terpenes and terpenoids in the fungus *Ganoderma lucidum* and *Ganoderma lingzhi.*

**Table 1 jof-06-00269-t001:** Major medicinal mushrooms and their main distributions.

MM Species	Common Name	Taxonomy	Geographic/Ecological Distribution
*Agrocybe aegerita*	Black Poplar mushroom	Basidiomycota Agaricomycetes Agaricales Bolbitiaceae	North temperate and subtropical zone
*Agaricus bisporus*	Button mushroom, Portobello mushroom, Common mushroom	Basidiomycota Agaricomycetes Agaricales Agaricaceae	USA, China, France, Netherlands, United Kingdom, Italy, Poland, Spain, Germany, Canada, Ireland, Belgium, Indonesia, Hungary and Mexico
*Agaricus blazei* (syn. *Agaricus brasiliensis*)	Royal Sun Agaricus, Almond Portobello	Basidiomycota Agaricomycetes Agaricales Agaricaceae	America, Brasil, Japan, China
*Amanita phalloides*	Death Cap	Basidiomycota Agaricomycetes Agaricales Amanitaceae	Europe, North American, Asia
*Boletus edulis*	Cep, Porcini, Penny Bun Bolete	Basidiomycota Agaricomycetes Boletales Boletaceae	China, Italy, France, Swiss, Germany
*Boletus speciosus*	Red-Capped Butter Bolete	Basidiomycota Agaricomycetes Boletales Boletaceae	Eastern North America, Southwest of China and Europe
*Chroogomphus rutilus*	Copper Spike	Basidiomycota Agaricomycetes Boletales Gomphidiaceae	China
*Clitocybe nebularis*	Clouded Funnel	Basidiomycota Agaricomycetes Agaricales Tricholomataceae	China, Japan, Taiwan, Europe, North America, North Africa
*Cryptoporus volvatus*	Veiled Polypore	Basidiomycota Agaricomycetes Polyporales Polyporaceae	Trunks of pine, fir and spruce
*Dichomitus squalens*	Common White-Rot fungus	Basidiomycota Agaricomycetes Polyporales Polyporaceae	Trunks of conifers such as pine and larch
*Flammulina velutipes*	Golden Needle mushroom	Basidiomycota Agaricomycetes Agaricales Physalacriaceae	Subtropical zone such as Japan, Russia, Australia and other countries as well as Europe, North America
*Floccularia luteovirens* (syn. *Armillaria luteovirens*)	Scaly Yellow mushroom	Basidiomycota Agaricomycetes Agaricales Tricholomataceae	Meadow at altitudes of 3000–4000 m above sea level
*Ganoderma atrum*	Black Ling-zhi	Basidiomycota Agaricomycetes Polyporales Polyporaceae	Tropical regions
*Ganoderma capense*	Dark Ling-zhi	Basidiomycota Agaricomycetes Polyporales Polyporaceae	Tropical regions
*Ganoderma japonicum*	Bloody Ling-zhi	Basidiomycota Agaricomycetes Polyporales Polyporaceae	Majority in tropical and subtropical regions of Asia, Australia, Africa and America, minority in temperate zone
*Ganoderma lucidum*	Reitake, Ling-zhi, Spirit Plant Reishi	Basidiomycota Agaricomycetes Polyporales Polyporaceae	Majority in tropical and subtropical regions of Asia, Australia, Africa and America, minority in temperate zone
*Ganoderma microsporum*	Small-Spored Ling-zhi	Basidiomycota Agaricomycetes Polyporales Polyporaceae	Subtropics zone
*Ganoderma lingzhi*	Ling-zhi	Basidiomycota Agaricomycetes Polyporales Polyporaceae	China, North Korea, Japan
*Ganoderma sinensis*	Zi-zhi	Basidiomycota Agaricomycetes Polyporales Polyporaceae	China, North Korea, Japan
*Ganoderma tsugae*	Hemlock Varnish Shelf	Basidiomycota Agaricomycetes Polyporales Polyporaceae	Northern and Montaine zone
*Grifola frondosa*	Maitake Hen of the Woods	Basidiomycota Agaricomycetes Polyporales Grifolaceae	Japan, China
*Hericium erinaceus*	Lion’s Mane mushroom, Bearded Tooth mushroom, Monkey-Head mushroom	Basidiomycota Agaricomycetes Russulales Hericiaceae	Broad-leaved forest or coniferous and broad-leaved mixed forest in northern temperate zone such as Western Europe, North America, China, Japan, Russia
*Inonotus obliquus*	Clinker Polypore, Birch Conk, Chaga	Basidiomycota Agaricomycetes Hymenochaetacles Hymenochaetaceae	Russia, China
*Lentinula edodes*	Shiitake, Black Forest mushroom, Golden Oak mushroom	Basidiomycota Agaricomycetes Agaricales Omphalotaceae	Distributed in an arc area on the west side of the Pacific Ocean, Japan, Papua New Guinea, Nepal, the Mediterranean coast and northern Africa
*Lignosus rhinocerotis*	Tiger Milk mushroom	Basidiomycota Agaricomycetes Polyporales Polyporaceae	China, Indonesia, Philippines, Sri Lanka, Australia, Thailand, Malaysia, Papua New Guinea and rainforests of East Africa
*Leucocalocybe mongolica* (syn. *Tricholoma mongolicum*)	Mongolia mushroom	Basidiomycota Basidiomycetes Agaricales Agaricales incertae sedis	Inner Mongolia in China
*Marasmius oreades*	Fairy Ring mushroom	Basidiomycota Agaricomycetes Agaricales Marasmiaceae	North America and Asia
*Morchella esculenta*	Common Morel, Yellow Morel, Sponge Morel	Ascomycota Pezizomycetes Pezizales Morohellaceae	Widely cultured over the world such as France, Germany, America, India, China, Russia, Sweden, Mexico, Spain, Czechoslovakia and Pakistan
*Morchella conica*	Black Morel, Sponge mushroom	Ascomycota Pezizomycetes Pezizales Morohellaceae	Distributed under broad-leaved forest, coniferous broad-leaved mixed forest, forest edge open space and weeds
*Naematelia aurantialba* (syn. *Tremella aurantialba*)	Golden Tremella	Basidiomycota Tremellomycetes Tremellales Naemateliaceae	Mountain forest of quercus, mutualism with *Stereum* spp.
*Ophiocordyceps sinensis*	Caterpillar fungus, Himalaya Viagra	Ascomycota Sordariomycetes Hypocreales Ophiocordycipitaceae	Southwest China, Nepal
*Pholiota adiposa*	Chestnut mushroom	Basidiomycota Agaricomycetes Agaricales Strophariaceae	Distributed on the dead willows in the forest in China
*Pleurotus citrinopileatus*	Golden Oyster mushroom, Tamogitake	Basidiomycota Agaricomycetes Agaricales Pleurotaceae	Widely cultured all over the world
*Pleurotus ostreatus*	Oyster mushroom	Basidiomycota Agaricomycetes Agaricales Pleurotaceae	Widely cultured all over the world
*Cerioporus squamosus* (syn. *Polyporus squamosus*)	Dryad’s Saddle, Pheasant’s Back mushroom	Basidiomycota Agaricomycetes Polyporales Polyporaceae	Widely distributed in hardwood forest of North America, Australia, Asia and Europe
*Poria cocos*	Fuling, China Root	Basidiomycota Agaricomycetes Polyporales Laetiporaceae	Parasitic on the roots of Pinaceae plants, mainly distributed in China
*Rhodonia placenta* (syn. *Postia placenta*)	Rosy Crust	Basidiomycota Agaricomycetes Polyporales Dacryobolaceae	Widely distributed all over the world
*Pseudosperma umbrinellum* (syn. *Inocybe umbrinella*)	Fibrous Hat	Basidiomycota Agaricomycetes Agaricales Inocybaceae	France
*Russula delica*	Milk-White Brittlegill	Basidiomycota Agaricomycetes Russulales Russulaceae	Taiga forest and mixed forests
*Russula lepida*	Rosy Russula	Basidiomycota Agaricomycetes Russulales Russulaceae	Widely distributed all over the world
*Sarcodon aspratus*	Black Tiger Paw	Basidiomycota Agaricomycetes Thelephorales Thelephoraceae	Southwest of China
*Schizophyllum commune*	Split Gill	Basidiomycota Agaricomycetes Agaricales Schizophyllaceae	Widely distributed all over the world
*Stropharia rugosoannulata*	Wine Cap Stropharia, Garden Giant, Burgundy mushroom, King Stropharia	Basidiomycota Agaricomycetes Agaricales Strophariaceae	Europe, North America, Asia
*Taiwanofungus camphoratus* (syn. *Antrodia camphorate*)	Poroid Brown-rot fungus, Stout Camphor fungus	Basidiomycota Agaricomycetes Polyporales incertae sedis	Mountain forest in Taiwan with altitudes of 450–2000 m
*Trametes versicolor* (syn. *Polystictus versicolor*)	Turkey Tail fungus	Basidiomycota Agaricomycetes Polyporales Polyporaceae	Global distribution; Broad-leaf woods
*Tropicoporus linteus* (syn. *Phelllinus linteus*)	Mesima, Black Hoof fungus	Basidiomycota Agaricomycetes Hymenochaetales Hymenochaetaceae	Distributed on the dead trees and trunks in China
*Xerocomellus chrysenteron* (syn. *Xerocomus Chrysenteron*)	Red Cracking Bolete	Basidiomycota Agaricomycetes Agaricales Agaricales incertae sedis	China
*Xylaria hypoxylon*	Candlestick fungus, Candlesnuff fungus, Carbon Antlers, Stag’s Horn fungus	Ascomycota Sordariomycetes Xylariales Xylariaceae	Northern Europe
*Xylaria nigripes*	Dead Moll’s Fingers	Ascomycota Sordariomycetes Xylariales Xylariaceae	China, mutualism with white ant
*Volvariella volvacea*	Straw mushroom	Basidiomycota Basidiomycetes Agaricales Pluteaceae	China, East Asia, Southeast Asia

**Table 6 jof-06-00269-t006:** Major genomic features of representative medicinal mushrooms.

Medicinal Mushroom	Genome Size Mb	Number of Genes	GC Content (%)	Known Genes Related to Immunomodulatory Effects	Genetic Manipulations (Transformation Method)	Refs
*Agrocybe aegerita*	44.7908	14110	49.2		Polyethylene glycol–mediated transformation (PEG)	[169]
*Agaricus bisporus*	30.78	10863	46.5		PEG, Electroporation, Particle bombardment, *Agrobacterium tumefaciens*-mediated transformation (ATMT)	[170,171,172,173]
*Flammulina velutipes*	35.64		49.76	*Fip-fve*	PEG, Electroporation, Electro-injection, Restriction enzyme-mediated integration (REMI), ATMT	[173,174,175,176,177]
*Ganoderma atrum*				*fip-gat*		[178]
*Ganoderma lucidum*	43.68		55.4	*fip-glu*, Mevalonate (MVA) pathway genes: *AACT* (acetyl-CoA acetyltransferase); *HMGS* (3-hydroxy-3-methylglutaryl-CoA synthase), *HMGR* (3-hydroxy-3-methylglutaryl-CoA reductase), *MVK* (mevalonate kinase), *MPK* (phosphomevalonate kinase), *MVD* (pyrophosphomevalonate decarboxylase), *IDI* (isopentenyl-diphosphate isomerase), *GPPs* (geranyl diphosphate synthase), *FPPs* (farnesyl diphosphate synthase), *SQS* (squalene synthase), *SE* (squalene monooxygenase), *OSC* (2,3-oxidosqualene-lanosterol cyclase), *P450* (cytochrome P450), *UGTs* (uridine diphosphate glycosyltransferases)	PEG, Electroporation, REMI	[173,179,180,181]
*Ganoderma sinensis*	48.96	15478	55.6	*fip-gsi*		[103,182]
*Ganoderma tsugae*	45.5			*fip-gts*		[183]
*Hericium erinaceus* (syn. *Hericium erinaceum*)	41.21		52.43		ATMT	[184,185]
*Lentinula edodes*	39.92	12051	46		PEG, Electro-injection, REMI, ATMT	[173,186]
*Pleurotus ostreatus*	34.36	12296	50.76		PEG, Electroporation, REMI, Particle bombardment	[173,187,188,189]
*Postia placenta*	66.6724	12716	*47.2*	*fip-ppl*		[108,190]
*Trametes versicolor*	44.794	14572	57.3	*fip-tvc*		[191,192]
*Volvariella volvacea*	35.72		*48.8*	*fip-vvo*	PEG, Particle bombardment, ATMT	[193,194]

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
