# Peer review of "Immunomodulatory Effects of Edible and Medicinal Mushrooms and Their Bioactive Immunoregulatory Products"

_jof, 2020, doi:10.3390/jof6040269_

Round 1
Reviewer 1 Report
The manuscript by Zhao and colleagues, summarizes the current state of the art about the use of medicinal mushrooms as natural products aimed to improve human health. The review is well written, and does not deal only with the medical use of medicinal mushrooms, but critically investigates the beneficial effects of their compounds. In particular, the authors focused on the immunomodulatory effects of these edible mushrooms.
The abstract is well written and clearly summarizes the aim of the review. However, I would suggest to the authors to add some additional keywords to make more attractive their manuscript.
The Introduction begins with a good description of the state of the art related to immunomodulators, and is focused on the current interest for medicinal mushrooms, especially those used as potential products to supply the human diet. As the authors reported in the introduction, the potential beneficial effects derived from the intake of medicinal mushrooms is correlated to the content of amino acids, salts and bioactive compounds present in these fungi. In particular, their bioactive compounds are specific for fungi and are not produced by other organisms such as plants.
I would suggest to the authors to add some references in the Introduction section for some key sentences (Line 31-34; Line 34-35; Line 40-42). Moreover, the authors should explain a bit more the increased interest in the use of medicinal mushrooms or plant-based products. For example, they should underline the increased interest of consumers in natural-based products with potential pharmacological and beneficial effects on human health (REF: https://doi.org/10.3390/nu12040992; https://doi.org/10.3390/nu12092605; https://doi.org/10.3390/nu12092605). In particular, thanks to their natural origin, these products are perceived to be safe since not of synthetic origin.
The section 2 is poorly described. The current knowledge about the use of bioactive molecules from medicinal mushrooms in specific pathologies could be certainly commented in a better way. Moreover, also in this section, some key sentences need one or more references (Line 70-72; Line 72-75; Line 97). This problem is also present in section 3 (Line 116-119; Line 122; Line 127-129; Line 163-166; Line 186-188; Line 189-191), in section 5 (Line 238-250) and in section 6 (Line 278-281; 288-297; 303-304; 339-340; 356-360; 384-388).
In the tables and in the main text of the manuscript the use of coloured words, unless strictly necessary, should be avoided. For example, why in line 145 the text is blue? Please, change the color from blue to black.
Genus and species are not always reported in italic (Line 86-87; Line 90-93). The authors should check troughtout the text.
Line 109: I would change medical with medicinal
Line 147: Report PSPC as a full name
Line 187: Pyrophosphate not pyrophosphates
Author Response
Dear Reviewer,
Thanks for your comments and suggestions. We really appreciate your efforts and comments. Please see below our specific responses to each of your comments.
The abstract is well written and clearly summarizes the aim of the review. However, I would suggest to the authors to add some additional keywords to make more attractive their manuscript.
Response: Thank you for your suggestions. We have added several keywords reflective of the contents of the manuscript.
The Introduction begins with a good description of the state of the art related to immunomodulators, and is focused on the current interest for medicinal mushrooms, especially those used as potential products to supply the human diet. As the authors reported in the introduction, the potential beneficial effects derived from the intake of medicinal mushrooms is correlated to the content of amino acids, salts and bioactive compounds present in these fungi. In particular, their bioactive compounds are specific for fungi and are not produced by other organisms such as plants.
I would suggest to the authors to add some references in the Introduction section for some key sentences (Line 31-34; Line 34-35; Line 40-42). Moreover, the authors should explain a bit more the increased interest in the use of medicinal mushrooms or plant-based products. For example, they should underline the increased interest of consumers in natural-based products with potential pharmacological and beneficial effects on human health (REF: https://doi.org/10.3390/nu12040992; https://doi.org/10.3390/nu12092605; https://doi.org/10.3390/nu12092605). In particular, thanks to their natural origin, these products are perceived to be safe since not of synthetic origin.
Response: Thank you for your suggestion, we have explained the reasons and added the relevant references in the manuscript.
The section 2 is poorly described. The current knowledge about the use of bioactive molecules from medicinal mushrooms in specific pathologies could be certainly commented in a better way. Moreover, also in this section, some key sentences need one or more references (Line 70-72; Line 72-75; Line 97). This problem is also present in section 3 (Line 116-119; Line 122; Line 127-129; Line 163-166; Line 186-188; Line 189-191), in section 5 (Line 238-250) and in section 6 (Line 278-281; 288-297; 303-304; 339-340; 356-360; 384-388).
Response: Thank you for your suggestion. We have provided additional information about bioactive molecules and their relevance to specific pathologies. Indeed, most of the details, including the references, are provided in the Tables 2-5. We have emphasized this in the revised version.
In the tables and in the main text of the manuscript the use of coloured words, unless strictly necessary, should be avoided. For example, why in line 145 the text is blue? Please, change the color from blue to black.
Response: Thank you for your reminder. We have changed the color from blue to black.
Genus and species are not always reported in italic (Line 86-87; Line 90-93). The authors should check throughout the text.
Response: Thank you for your kind reminder. We have checked through the text and made sure that all genus and species names are italicized
Line 109: I would change medical with medicinal
Response: Thank you for your suggestion, we have changed the word medical to medicinal.
Line 147: Report PSPC as a full name
Response: Thank you for your reminder. We have supplied the full name of PSPC, polysaccharide-protein complex.
Line 187: Pyrophosphate not pyrophosphates
Response: Thank you for your suggestions, we have changed the word pyrophosphates to pyrophosphate in the manuscript.
Reviewer 2 Report
Thank you for your interesting research and your exhaustive work. There are some important points that need to be carefully revised:
- Medicinal mushrooms. Lines 86-93. Please, use Italic type for mushrooms species and for mushrooms genera.
- Medicinal mushrooms. Table 1. Is there any common name for empty spaces?
- Some relevant isolated polysaccharides are missing in this section. For instance, the linear α-1,3-glucan that was isolated from Lentinula edodes and exerted immune-modulatory properties. Please, utilize this useful reference: https://doi.org/10.1016/j.carbpol.2019.115521
- What about chitin (or derived chito-oligosaccharides)? Maybe some information should be added regarding immune responses to fungal chitin.
- What about European legislation regarding approved products? Maybe some information should be added to data from Japan.
- Figure 3. This is a relevant figure in your manuscript. Please, could you revise typographic mistakes?
- Maybe non desirable effects of lections should be mentioned too.
Author Response
Dear Reviewer,
Thank you for your comments and suggestions. We really appreciate your time and effort in trying to help us improve the manuscript. Please see below our specific responses to each of your comments/suggestions.
1. Medicinal mushrooms. Lines 86-93. Please, use Italic type for mushrooms species and for mushrooms genera.
Response: Thank you for your kind reminder. We have revised the formats of mushroom species and genera to ensure all Latin names are italicized.
2. Medicinal mushrooms. Table 1. Is there any common name for empty spaces?
Response: Done. We have searched extensively and filled the common names for all empty spaces.
3. Some relevant isolated polysaccharides are missing in this section. For instance, the linear α-1,3-glucan that was isolated from Lentinula edodes and exerted immune-modulatory properties. Please, utilize this useful reference: https://doi.org/10.1016/j.carbpol.2019.115521
Response: Thank you for your suggestion, we have supplied additional information on linear α-1,3-glucan isolated from Lentinula edodes in the manuscript and the reference is cited (#40).
4. What about chitin (or derived chito-oligosaccharides)? Maybe some information should be added regarding immune responses to fungal chitin.
Response: Thank you for your comment. We have searched for the references of fungal chitin and their immunomodulatory effects. Indeed, chitin and its derivatives chitosan and chito-oligosaccharides are very abundant in fungi, including in edible and medicinal mushrooms. Several fungal chitin, chitosan, and chito-oligosaccharides have shown promising benefits to humans and human health. For example, chitins have been used in plant protection and food processing; chitosan in diagnosis, drug delivery, infection control, molecular imaging, and wound healing; and chito-oligosaccharides in antimicrobial and antitumor activities. However, none of those analyzed chitin, chitosan, and chito-oligosaccharides for human effects have come from edible or medicinal mushrooms. This group of natural products from edible and medicinal mushrooms definitely represents an area of future development. We have provided these updates, including a few references in the Conclusions and Perspectives section.
5. What about European legislation regarding approved products? Maybe some information should be added to data from Japan.
Response: Thank you for your suggestion. We have added a very recent reference on the topic and briefly discussed the implications and challenges at the end of the paper. [Gründemann, C., Reinhardt, J.K., Lindequist, U. European medicinal mushrooms: Do they have potential for modern medicine? – An update. Phytomedicine. 2020, 66, 153131. https://doi.org/10.1016/j.phymed.2019.153131]
6. Figure 3. This is a relevant figure in your manuscript. Please, could you revise typographic mistakes?
Response: Thank you for your careful checking. We have revised the figure and corrected the typographic mistakes.
7. Maybe non desirable effects of lections should be mentioned too.
Response: Thank you for your suggestion, we have added a couple of references describing non-desirable effects of medicinal mushrooms and discussed their broad implications in the Conclusions and Perspectives section.
This manuscript is a resubmission of an earlier submission. The following is a list of the peer review reports and author responses from that submission.
Round 1
Reviewer 1 Report
The article is a very interesting compilation of information for the scientific community. However, there are many typographical errors, mistakes on taxonomic information and inconsistency on information provided that discourages and distracts the reviewers' interest. I started reading the article and performed verification of information on table 1, and had to stop because of inconsistencies and errors provided.
I recommend rejection until a through revision is done prior to resubmission.
here are some recommendations:
Line 84-90 scientific names need to be in italics
Table 1 -
Please separate better each one because the Taxonomy column gets too crowded and its confusing for people unfamiliar with these terms.
Agrocybe aegerita - common name: black poplar mushroom
Agaricus bisporus - common name: Please add, Common mushroom
Agaricus blazei (syn. Agaricus brasiliensis) Please eliminate Basidiomycetes to keep consistency of Phylum, Class, Order, Family that you provided in the table
Armillaria luteovirens correct name to Floccularia luteovirens, common name: scaly yellow mushroom
Antrodia camphorate Please change to Taiwanofungus camphoratus the correct genus is Taiwanofungus, eliminate Polyporales as family because there is not an official denomination (M. Zang & C.H. Su) Sheng H. Wu, Z.H. Yu, Y.C. Dai & C.H. Su, 2004
Boletus edulis Common names: Cep, Porcini or Penny Bun Bolete and please change to Boletaceae is the correct family, the Boletoideae is the subfamily. Please be check all of the taxonomy and be consistent, they all need to be the same.
Boletus speciosus Change to Boletaceae is the correct family, the Boletoideae is the subfamily. Please be check all of the taxonomy and be consistent, they all need to be the same.
Chroogomphis rutilus Please fix the correct genus is Chroogomphus, Common name: Copper spike
Clitocybe nebularis common name Clouded funnel
Ophiocordyceps sinensis Taxonomy is completely wrong: Ascomycota, Sordariomycetes, Hypocreales, Ophiocordycipitaceae
Cryptoporus volvatus Common name veiled polypore, Taxonomy is completely wrong: Basidiomycta, Agaricomycetes, Polyporales, Polyporaceae
Please revise the taxonomy of all your mushrooms using the taxonomy browser from NCBI:
https://www.ncbi.nlm.nih.gov/Taxonomy/Browser/wwwtax.cgi
Ganoderma lucidum common name please add Reishi
Figure 1 Please be consistent in the legend and the figure, use scientific names in both places for consistency and ease of identifying the species being shown.
***stopped review***